# Relationship between self-efficacy, adversity quotient, COVID-19-related stress and academic performance among the undergraduate students: A protocol for a systematic review

**Ridho Roqwan Ikbar[1], Noh Amit**[2,3]*, **Ponnusamy Subramaniam**[2,4], **Norhayati Ibrahim**[2,4]

**1** Health Psychology Program, Faculty of Health Sciences, Universiti Kebangsaan Malaysia, Kuala Lumpur, Federal Territory of Kuala Lumpur, Malaysia, **2** Clinical Psychology and Behavioral Health Program, Faculty of Health Sciences, Universiti Kebangsaan Malaysia, Kuala Lumpur, Federal Territory of Kuala Lumpur, Malaysia, **3** Centre for Community Health Studies, Faculty of Health Sciences, Universiti Kebangsaan Malaysia, Kuala Lumpur, Federal Territory of Kuala Lumpur, Malaysia, **4** Center for Healthy Ageing and Wellness, Faculty of Health Sciences, Universiti Kebangsaan Malaysia, Kuala Lumpur, Federal Territory of Kuala Lumpur, Malaysia

* nohamit@ukm.edu.my

**Data Availability Statement:** No datasets were generated or analysed during the current study. All

## Abstract

### Background

This systematic review aims to review research manuscripts during the COVID-19 pandemic that focus on the relationship between self-efficacy, adversity quotient, COVID-19-related stress and academic performance on a range of undergraduate student.

### Methods

The authors will perform comprehensive searches of published studies in electronic databases such as PMC, PubMed, Scopus, Cochrane Library and Web of Science by using the following search terms: 'self-efficacy' AND 'adversity quotient' AND 'stress' AND 'academic performance' AND 'student' AND 'COVID-19 pandemic'. Only full-text articles in English language are included. Two reviewers will independently conduct the article selection, data extraction, and quality assessment. Any possible disagreement will be resolved by discussion, and one arbitrator (NA) will adjudicate unresolved disagreements.

### Results

This review will provide an updated overview of investigating the relationship between self-efficacy, adversity quotient, COVID-19-related stress and academic performance on a range of undergraduate student during the COVID-19 pandemic. Ultimately, based on this systematic review, we will recommend the direction for future research.

relevant data from this study will be made available upon study completion.

**Funding:** The author(s) received no specific funding for this work.

**Competing interests:** The authors have declared that no competing interests exist.

## Conclusion

The result of the study may help the researchers to find an updated overview of various studies in related topic.

## Ethics and dissemination

Data from published studies will be used. Therefore, ethical approval is not required prior to this systematic review. The results will be published in a peer-reviewed journal.

## Introduction

Generally, stress refers to a person's physical or emotional response to the demands or pressures of daily life [1]. Stress among students could be associated to various factors such as individual factors and environmental factors. The example of individual factors are physical condition, motivation, and personality type of the students themselves. In addition, environmental factors cover several aspects such as family, work, facilities, environment, lecturers, and others [2].

According to Gunawati, and Listiara [3], factors related to stress among students are namely internal factors and external factors. The internal factors include mindset, personality, and beliefs meanwhile the external factors cover heavy lessons workloads, pressure to achieve high academic excellence, social status encouragement, and parent competition with each other.

During the COVID-19 outbreak, a population-based survey showed post-crisis mental health among the students [4]. From this perspective, the academic context was affected by the lockdown worldwide. Indeed, The United Nations Educational, Scientific and Cultural Organization (UNESCO) reported that in consequence of the massive universities' closure, the scheduled activities and student's accommodations were suspended, all interactions were shifted to online platforms, leading to a major change in students' academic life [5].

In most cases, the literature has documented the negative influence of COVID-19 pandemic on the first-year undergraduate students' stress to academic. For instance, Sundarasen et al. [6] investigated the psychological impact of COVID-19 pandemic on university students in Malaysia. The stress was measured by using Zung's [7] self-rating anxiety scale (SAS). Out of 359 first-year undergraduate students studied, 93.6% reported low stress, 3.9% reported moderate stress, and 2.5% reported high stress. Furthermore, similar study was conducted by Putri and Ariana in Indonesian context where the stress level was measured by Student-life Stress Inventory [8]. The findings show that out of 252 first-year undergraduate students studied, 110 of them (43.6%) reported low stress, 89 respondents reported moderate stress (35.3%), and 53 respondents (21%) reported high stress.

According to Son, Hegde, Smith, Wang, and Sasangohar [9], there were stressors which affected the academic's life among students during COVID-19 pandemic. These are their own health and the health of loved ones, difficulty in concentration, sleeping habits, social relation/ social isolation, academic performance, financial difficulties, depressive thoughts, suicidal thoughts.

COVID-19 outbreak may enhance fear among the students such as their own health and their loved ones. They were worried about the family and relatives who were more susceptible to the virus such as older adults, those who have existing severe health problems, pregnant

women, and women who have just gave birth. Students were also concerned about their family whose jobs increased the risk of exposure to COVID-19 such as social and health care workers.

There are various changes in terms of educational approaches which potentially lead to difficulty in concentrating on academic work. Son et al. [9] found that most of the students (173; 89%) stated that their home is distractive. For them, home is more appropriate place to than to study (79; 46%). In fact, students were easier to be bothered by their family members at home. Other factors influencing students' concentration were social media, internet, and video games (19; 11%). Some students (18; 10%) mentioned that online classes were subject to diversion due to social interaction deficiency and continuous attention to computer screen. In addition, monotone life patterns mentioned by some students could negatively affect concentration on academic work (5; 3%).

Students reported that their sleep patterns were disrupted due to COVID-19 pandemic. Further, they mostly stated that they tended to stay up later or wake up later they did before the pandemic. It could be associated with the need to be having online meeting through online platforms which could be carried out anytime and anywhere. Another irritative impact brought by the COVID-19 pandemic was irregular sleep patterns such as inconsistent time to go to bed and to wake up from day to day.

Son et al. [9] stated that most students (167; 86%) answered that the COVID-19 pandemic increased the social isolation. Over half of students (91; 54%) stated that it limited their interactions with friends significantly. Some students (52; 31%) stated that they worried about a lack of in-person interactions. Others (9; 5%) mentioned that disturbance to their outdoor activities have affected their mental health.

Furthermore, in 2020, Son et al. [9] found that majority of students (159; 82%) concerned on their academic performance impacted by the COVID-19 pandemic. Fear of lower performance and delay in completion of studies are also the reasons to induce stress among students during COVID-19. The biggest challenge was the transition to classes through online platforms (61; 38%). In particular, students concerned on sudden changes in the syllabus, the quality of the classes, technical issues with online applications, and the difficulty of online learning. Some students (36; 23%) worried about progress in research and class projects because of social restrictions to keep social distancing and the lack of physical interactions with other students. Others (23; 14%) stated the uncertainty about their grades on the learning through online platforms to be a major stressor. In addition, others (12; 8%) indicated their lower motivation to learn and tendency to procrastinate.

Majority of students revealed their concerns about the financial situations which were impacted by COVID-19. Some students stated that COVID-19 has affected employment opportunities such as part-time jobs and internships. Others expressed the financial difficulties of their family members, mostly parents, who got laid off and got pay cut during the pandemic.

Regarding the impact of the COVID-19 pandemic, the study by Son et al. [9] found that 44% of students mentioned that they were having some depressive thoughts. Most students attributed the depressive thoughts to factors such as loneliness (33%), insecurity (12%), hopelessness (10%), academic performance concerns (8%), and overthinking (5%).

Out of 195 students, 16 students revealed that the pandemic led to suicidal thoughts with 5% experiencing these thoughts as low level and 3% as moderate level. The reasons were dealing to academic performance, problems with family as they returned home, and fear of insecurity and uncertainty.

Lazarus and Folkman [10] defined that psychological stress is a relationship between an individual and the environment which is valued by the way people exceed their sources and endanger their well-being. This relationship is going through two significant phases that are

cognitive appraisals and coping. The cognitive appraisal is defined as "the process of categorizing an encounter, and its various facets, with respect to its significance for well-being" (Lazarus and Folkman [10], p. 31). Furthermore, cognitive appraisal covers primary appraisal and secondary appraisal.

Primary appraisal manages to answer "Am I trouble or being benefited, now or in the future, and in what ways?". When people answer "yes", it means that the situation can be classified as threat, challenge or loss. Loss suggests harms occurred meanwhile threat and challenge can be meant past experiences. Threat refers to physical and psychological potential while challenge means that someone's concern on his achievement, reward, and growth; however, this threat and challenge are not correspondent with. Lazarus and Folkman [10] mentioned that threat and challenge appraisals are not two ends of a single continuum. For example, Folkman and Lazarus [10] showed that students waiting for an exam evaluate it as threatening and challenging event. Secondary appraisal refers to an assessment of coping resources. It has an attempt to answer "Can I cope with this situation?". It reflects someone's ability to deal with the situation because one has the resources (physical, social, psychological, or material).

Lazarus and Folkman [10] proposed that coping has two main functions. First is to regulate emotions or distresses in the stressful situation (emotion-focused coping). The second one is to manage the problem causing the stress by directly changing the elements of the stressful encounter (problem-focused coping). Even though both are used in most stressful situation, they are nevertheless dependent in terms of the way one appraises the situation and of the antecedents of the model.

Students who experience stressful conditions will have a negative impact on the performance. For instance, when the students experience prolong stress, the stress may affect their academic achievement and lead into loss of enthusiasm which could be associated with low GPA [11].

Oboth and Odiemo [12] found that that only 35.6% of students experiencing low level of stress while the rest 64.4% were experiencing moderate to high levels of stress. The result showed that stress level and academic performance had a significant relationship. Regression analysis showed that the higher the level of stress, the lower the academic performance among the students. This finding is consistent with other studies which were conducted in Malaysia and Indonesia. In Malaysia, Elias, Ping, and Abdullah [13] study which was conducted at Universiti Putra Malaysia, revealed that stress is shown to be significantly correlated to academic performance. On the other hand, in Indonesia, in Tumonggor, Sutanti, Evan, Winata [14] study which was conducted among the students of Krida Wacana Christian University class of 2017 to 2019 during the COVID-19 pandemic, shown that there is a relationship between stress and student's academic performance.

Furthermore, to define protective factors of stress among the students, some studies were initiated. Majority of researches investigated the relationship between self-efficacy and adversity quotient. Somaratne, Jayawardena, and Perera [15] found that level of adversity quotient was significantly related to the level of stress indicating that the higher level of adversity quotient, the lower level of stress. By using Pearson Product-Moment correlation analysis, the results showed that there was a strong negative relationship between perceived stress and all the sub-dimensions of AQ (control, ownership, reach, endurance). By having higher levels of control, students could be proactively dealing with adverse events and are competent to turn obstacles into opportunity. The negative relationship of ownership indicates that students with high AQ levels tend to feel liable to improve the obstacles and face them responsibly. Besides that, the negative relationship between reach and stress indicates that the respondents with high AQ levels do not let obstacles to reach other areas of life that restrict the negative impacts

to that particular event. Similarly, as implied by the negative relationship between endurance and stress, the respondents with high AQ tend to find solutions to overcome the obstacles.

On the other hand, Lee, Kim and Wachholtz [16] found that perceived stress was negatively associated with both self-efficacy among the 279 undergraduate students from three universities in Seoul, and Dae Jeon Korea. In terms of gender distribution, 188 (67.4%) of the participants were female and 91 (32.6%) students were male. It was reported that that self-efficacy and stress among students showed a negative significant relationship indicating also that the higher self-efficacy, the lower stress level of individual. This means if students were confident in their ability to meet the challenges, they would experience positive interpretation of current situation, leading to enhance the chance to avoid having stress.

The rational of conducting current systematic review on the relationship between self-efficacy, adversity quotient, COVID-19-related stress and academic performance is to enhance knowledge on stress among the students during COVID-19 pandemic and to examine the methodological approach in researching student's stress during the COVID-19 pandemic.

Hence, the present systematic review aims to review research manuscripts during the COVID-19 pandemic which focus on the relationship between self-efficacy, adversity quotient, COVID-19-related stress and academic performance on a range of undergraduate student. Ultimately, based on this systematic review, we will recommend the direction for future research.

## Materials and methods

### Study protocol registration

The study protocol of this review has been registered in the International Prospective Register of Systematic Reviews/PROSPERO (registration number: CRD42022336391).

### Systematic review protocol

The PRISMA guidelines and the PRSIMA flowchart will be adapted to summarize the search process. PRISMA is the revised version of the Quality of Reporting of Meta-analyses (QUAROM) guideline, consisting of a PRISMA-P checklist [17] and PRISMA flowchart [18].

### Eligibility criteria

This systematic review focuses on the published journal articles which discuss the relationship between self-efficacy, adversity quotient, COVID-19-related stress, and academic performance among the undergraduate students.

Inclusion criteria for the systematic review are study design and language. In term of study design, the study variable should cover two or more variable such as self-efficacy, COVID-19-related stress, adversity quotient and academic performance. It should cover published full-text English articles. Furthermore, the review also will include studies examining the undergraduate student population during the COVID-19 pandemic.

There four exclusion criteria in this study. The first is the studies which do not discuss the relationship between self-efficacy, adversity quotient, COVID-19-related stress, and academic performance. The second is the studies which do not focus on undergraduate students. The third is the studies which do not define and discuss the variables during the COVID-19 pandemic. The last is it excludes unpublished books, thesis or dissertations.

The primary outcome of interest is the relationship between self-efficacy, adversity quotient, COVID-19-related stress and academic performance among the undergraduate students during the COVID-19 pandemic. Outcomes will be reported.

## Information sources

Literature search strategies will be carried out by using five databases. These are the PubMed Central (PMC), PubMed, Scopus, Cochrane Library and Web of Science databases and published between December 2019 until August 2022. The literature search will be limited to the English language and student subjects. The keywords for the search are: 'self-efficacy' AND 'adversity quotient' AND 'stress' AND 'academic performance' AND 'student' AND 'COVID-19 pandemic'.

## Search strategy

An example for the search strategy is the writer when searching article in PubMed will the keyword 'self-efficacy' AND 'adversity quotient' AND 'stress' AND 'academic performance' AND 'student' AND 'COVID-19 pandemic'. The time frame of the articles is December 2019 – August 2022. This time frame reflects the emergence COVID-19 and the latest information of the pandemic. One reviewer will select articles against the inclusion criteria based on their title, abstract and full-text copies. Each accepted full-text manuscript will be analyzed by exploring their methodology such as study design, sampling method, sample size, psychological tools used, and their result on the relationship between self-efficacy, adversity quotient, COVID-19-related stress, and academic performance among the undergraduate students.

All selected articles will be obtained for data synthesis. The study selection will be done in concordance with the flow chart for systematic review (Fig 1). However, disparities in reviewer selections will be resolved at a meeting between reviewers prior to selected articles being retrieved.

Only quantitative studies will be sought. No study design limits will be imposed on the search, but there will be time or language limits although only studies during the COVID-19 pandemic and in English will be included.

## Study records

Literature search results, citation abstracts, and full-text articles will be shared via email that facilitates collaboration among reviewers during the study review process. The reviewers will review the manuscripts based on the inclusion and exclusion criteria.

The reviewers will screen the titles and abstracts resulted by the search against the inclusion criteria. In addition, the reviewers will obtain full-text reports for all titles that appear and decide whether these meet the inclusion criteria and record the reasons for excluding trials.

The team of reviewers will extract data and in duplicate from each eligible study. Data abstracted will include demographic information, methodology, and all reported important outcomes. The reviewers will resolve disagreements by discussion, and one arbitrator (NA) will adjudicate unresolved disagreements.

## Data extraction

Each accepted manuscript for review will be analyzed through a systematic and careful process. The full text of the articles will be read, exploring their methodology and results. Information on the study's design, sampling method, sample size, psychological tools used, and operational definitions will be recorded. A risk of bias analysis will be carried out for each individual study. The summary of the PRISMA checklist and a sample for electronic search strategy for the present systematic review will be presented accordingly.

## Data items

Some key terms are clarified in this study. Firstly, Pajares and Miller [19] assume that self-efficacy refers to student's beliefs in their ability to master new skills and tasks. Secondly,

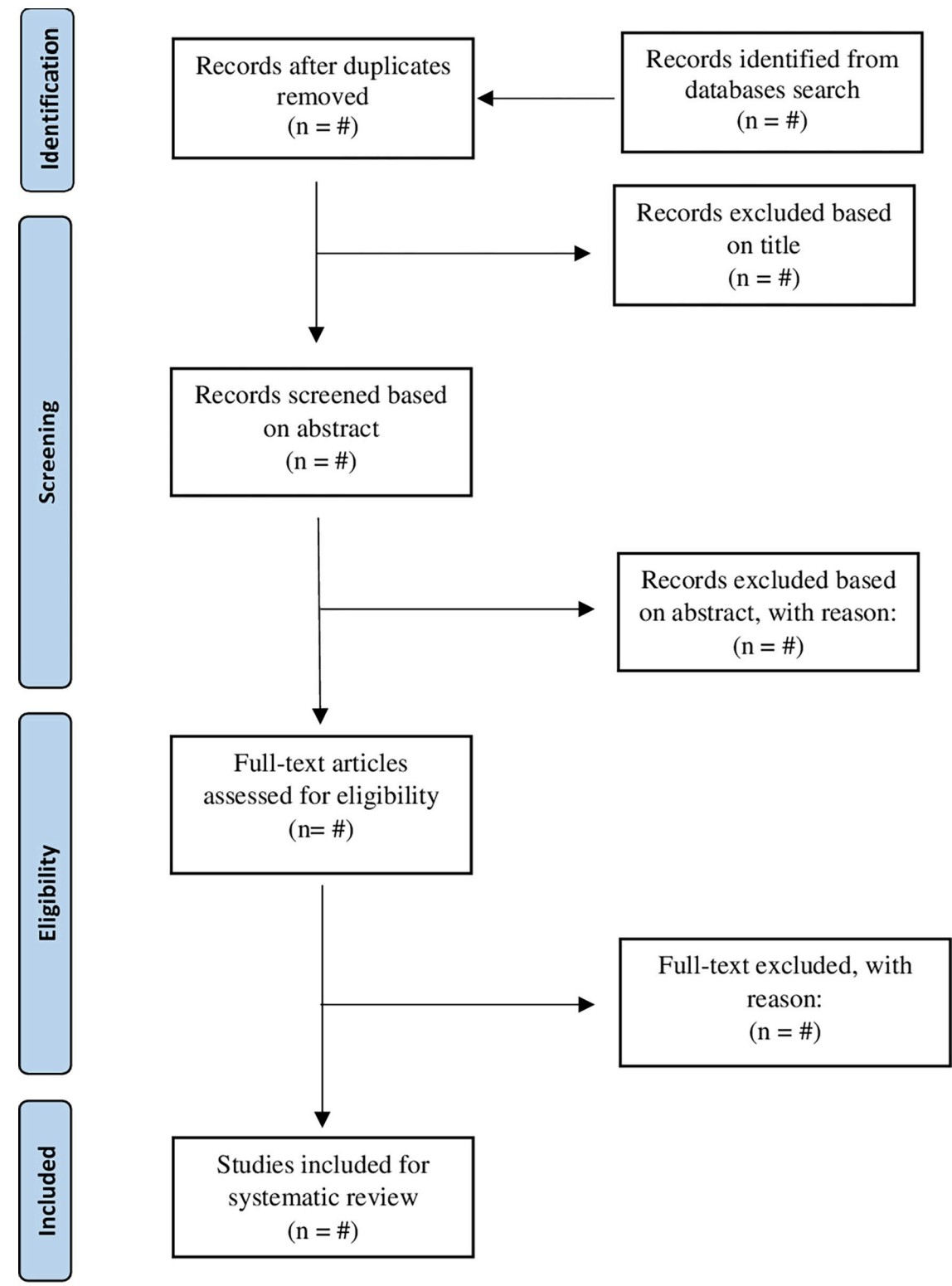

**Fig 1. PRISMA flow chart.**

Adversity Quotient (AQ) is defined as the ability to surmount life's adversities, and to turn every challenge into opportunities for personal success [20]. Thirdly, stress refers to a person's physical or emotional response to the demands or pressures of daily life [21]. In academic background, stress pervades the life of students, and tends to affect their mental and physical health, as well as their ability to perform schoolwork effectively [22]. Lastly, academic performance defined by grade point average (GPA). In addition, undergraduate students refer to students at a college or university who has not received a first and especially a bachelor's degree [23].

By using a systematic narrative synthesis [24], we will review the reported result of the study on the relationship between self-efficacy, adversity quotient, COVID-19-related stress and academic performance among the undergraduate students during COVID-19 pandemic. It is possible to extract additional results such as demographic factors differences on age, gender, length of study, study program, etc.

## Outcomes and prioritization

The primary outcome will be the relationship between self-efficacy, adversity quotient, COVID-19-related stress and academic performance among the undergraduate students during COVID-19 pandemic.

## Risk of bias

Systematic reviews aim to collate and synthesise all studies that meet prespecified eligibility criteria using methods that attempt to minimise bias [25]. To get reliable conclusions, review authors must carefully consider that the potential limitations of the included studies (its design, conduct, analysis, and presentation) are appropriate to answer its research question. Many tools for assessing the quality of the study such as the risk of bias assessment tool. In this systematic review, the risk of bias of each individual study is determined using two tools, they are the National Heart, Lung, and Blood Institute Quality Assessment Tool for Observational Cohort and Cross-Sectional Studies [26] and The Cochrane Collaboration's tool for assessing risk of bias [27]. These guidelines will be used in the reviews at outcomes level. The information from this risk of bias assessment aims to evaluate the quality of the research manuscripts included in the present review. Regardless of the manuscript's evaluation, its strengths and weaknesses are used to generate methods to enhance future studies. These judgements will be made by two reviewers based on the criteria for judging the risk of bias. Disagreements will be resolved first by discussion and then by consulting the third and fourth author for arbitration.

## Data synthesis

A systematic narrative synthesis will be provided with information presented in the text and tables to summarize and explain the characteristics and findings of the included studies. The reviewers will use narrative synthesis to explore the relationship between two or more variables of self-efficacy, adversity quotient, COVID-19-related stress and academic performance and other additional findings within the included studies. The summary of the PRISMA checklist and a sample for electronic search strategy for the present systematic review will also be presented respectively.

## Discussion

The systematic review focuses on the published journal articles discussing the relationship between self-efficacy, adversity quotient, COVID-19-related stress, and academic performance

among undergraduate students. It should cover published full-text English articles and studies which examine the undergraduate student population during the COVID-19 pandemic. Five databases will be used, namely the PubMed Central (PMC), PubMed, Scopus, Cochrane Library and Web of Science databases. The articles which will be selected for this study are the articles published between December 2019 until August 2022.

Reviewers will select articles against the inclusion criteria based on their title, abstract and full-text copies. Each accepted full-text manuscript will be analyzed by using narrative synthesis to explore their methodology such as study design, sampling method, sample size, psychological tools used, and their result on the relationship between self-efficacy, adversity quotient, COVID-19-related stress, and academic performance among the undergraduate students during the COVID-19 pandemic. The data supporting the findings of this study will be made available within the article and/or its supplementary materials.

To reach the reliable conclusions, the risk of bias of each individual study is determined using two tools. The first is the National Heart, Lung, and Blood Institute Quality Assessment Tool for Observational Cohort and Cross-Sectional Studies and the second is The Cochrane Collaboration's tool for assessing risk of bias.

## Expected limitation

The findings of this review may be restricted by a limitation. The review process inevitably identifies studies that are different in their design, quality of methodology, specific interventions used, also educational background and types of respondents studied. There is potential subjectivity when deciding how similar studies must be.

## Supporting information

**S1 Table. PRISMA-P checklist.**
(PDF)

## Author Contributions

**Conceptualization:** Ridho Roqwan Ikbar, Noh Amit.

**Data curation:** Ridho Roqwan Ikbar.

**Formal analysis:** Ridho Roqwan Ikbar, Noh Amit.

**Methodology:** Ridho Roqwan Ikbar, Noh Amit.

**Project administration:** Ridho Roqwan Ikbar, Noh Amit.

**Resources:** Ridho Roqwan Ikbar.

**Supervision:** Noh Amit, Ponnusamy Subramaniam, Norhayati Ibrahim.

**Validation:** Noh Amit.

**Writing – original draft:** Ridho Roqwan Ikbar, Noh Amit.

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
