## [Decision Letter · Decision Letter 0]

1 Aug 2022

PONE-D-22-14527Relationship between Self-Efficacy, Adversity Quotient, COVID-19-related Stress and Academic Performance among the Undergraduate Students: A Systematic Review A protocol for a systematic review

PLOS ONE

Dear Dr. Amit,

Thank you for submitting your manuscript to PLOS ONE. After careful consideration, we feel that it has merit but does not fully meet PLOS ONE’s publication criteria as it currently stands. Therefore, we invite you to submit a revised version of the manuscript that addresses the points raised during the review process.

We look forward to receiving your revised manuscript.

Kind regards,

Muhammad Shahzad Aslam, Ph.D.,M.Phil., Pharm-D

Academic Editor

PLOS ONE

Journal Requirements:

2. PLOS ONE does not copy edit accepted manuscripts (https://journals.plos.org/plosone/s/criteria-for-publication#loc-5). To that effect, please ensure that your submission is free of typos and grammatical errors. 

5. Thank you for submitting the above manuscript to PLOS ONE. During our internal evaluation of the manuscript, we found significant text overlap between your submission and the following previously published works, some of which you are an author.

-   https://www.macrothink.org/journal/index.php/ijld/article/view/13840

-   https://www.frontiersin.org/articles/10.3389/fpsyg.2020.576758/full

The text that needs to be addressed involves the Introduction and Discussion sections.

Please revise the manuscript to rephrase the duplicated text, cite your sources, and provide details as to how the current manuscript advances on previous work. Please note that further consideration is dependent on the submission of a manuscript that addresses these concerns about the overlap in text with published work.

Additional Editor Comments:

1-Please provide background of the study

2-Please provide a literature review

3-Please provide a theoretical foundation of the study.

Reviewers' comments:

Reviewer's Responses to Questions

**Comments to the Author**

1. Does the manuscript provide a valid rationale for the proposed study, with clearly identified and justified research questions?

Reviewer #1: Yes

Reviewer #2: Yes

2. Is the protocol technically sound and planned in a manner that will lead to a meaningful outcome and allow testing the stated hypotheses?

Reviewer #1: Yes

Reviewer #2: Yes

3. Is the methodology feasible and described in sufficient detail to allow the work to be replicable?

Reviewer #1: Yes

Reviewer #2: Yes

4. Have the authors described where all data underlying the findings will be made available when the study is complete?

Reviewer #1: Yes

Reviewer #2: No

5. Is the manuscript presented in an intelligible fashion and written in standard English?

Reviewer #1: Yes

Reviewer #2: Yes

6. Review Comments to the Author

You may also provide optional suggestions and comments to authors that they might find helpful in planning their study.

Reviewer #1: Your study is quite reasonable, intellectual and informative but in this review article many time every statement or heading start with " This review article" , so instead of this statement use suitable word. overall this manuscript is accepted

Reviewer #2: 1. Include Cochrane library in Information source

2. Include months with years in Information source

3. Please provide details list of keywords with Boolean expression

4. Please provide detail and separate heading of data extraction methods.

5.Please cite literature properly where necessary such as a systematic narrative synthesis. Cite reference.

6. Please provide operational definition of variables.

7.Please explain or define what does undergraduate students means? Detail operational information is mandatory.

8. Justify tool using for Bias assessment? Please add Cochrane bias tool or some more tool and compare your findings in actual study.

9. Please discuss the methodology of your proposed study and mention under the heading of discussion?

10. What will be expected limitation of the study?

7. PLOS authors have the option to publish the peer review history of their article (what does this mean?). If published, this will include your full peer review and any attached files.

Reviewer #1: No

Reviewer #2: **Yes: **Saima Nisar

---

## [Author Response · Author response to Decision Letter 0]

14 Sep 2022

Journal Requirements:

File naming was edited to comply with the style requirements. 

2. PLOS ONE does not copy edit accepted manuscripts.

The manuscript has been submitted for proofreading by an English language expert.

The title in the submission form has been revised as follow (please refer to the page 1)

The revised title: Relationship between self-efficacy, adversity quotient, covid-19-related stress and academic performance among the undergraduate students: a protocol for a systematic review.

4. Please include captions for your Supporting Information files at the end of your manuscript, and update any in-text citations to match accordingly.

The captions for Supporting Information are amended. The changes are described on lines 414-415.

“S1 Fig. PRISMA Flow Chart.”

“S1 Table. PRISMA-P Checklist.”

5. Thank you for submitting the above manuscript to PLOS ONE. During our internal evaluation of the manuscript, we found significant text overlap between your submission and the following previously published works.

The sentences were rephrased. 

The revision could be found on lines 57-60 and 153-157.

Insertion:

57-60: Indeed, The United Nations Educational, Scientific and Cultural Organization (UNESCO) reported that in consequence of the massive universities’ closure, the scheduled activities and student’s accommodations were suspended, all interactions were shifted to online platforms, leading to a major change in students’ academic life [5].

154-158: Oboth and Odiemo [12] found that that only 35.6% of students experiencing low level of stress while the rest 64.4% were experiencing moderate to high levels of stress. The result showed that stress level and academic performance had a significant relationship within age and gender groups. Regression analysis showed that the higher the level of stress, the lower the academic performance among the students.

Academic Editor:

1. Please provide background of the study.

The background was provided on lines 44-71.

Revision:

Generally, stress refers to a person’s physical or emotional response to the demands or pressures of daily life [1]. Stress among students could be associated to various factors such as individual factors and environmental factors. The example of individual factors are physical condition, motivation, and personality type of the students themselves. In addition, environmental factors cover several aspects such as family, work, facilities, environment, lecturers, and others [2].

According to Gunawati, and Listiara [3], factors related to stress among students are namely internal factors and external factors. The internal factors include mindset, personality, and beliefs meanwhile the external factors cover heavy lessons workloads, pressure to achieve high academic excellence, social status encouragement, and parent competition with each other.

During the COVID-19 outbreak, a population-based survey showed post-crisis mental health among the students [4]. From this perspective, the academic context was aﬀected by the lockdown worldwide. Indeed, The United Nations Educational, Scientific and Cultural Organization (UNESCO) reported that in consequence of the massive universities’ closure, the scheduled activities and student’s accommodations were suspended, all interactions were shifted to online platforms, leading to a major change in students’ academic life [5].

In most cases, the literature has documented the negative influence of COVID-19 pandemic on the first-year undergraduate students’ stress to academic. For instance, Sundarasen et al. [6] investigated the psychological impact of COVID-19 pandemic on university students in Malaysia. The stress was measured by using Zung’s [7] self-rating anxiety scale (SAS). Out of 359 first-year undergraduate students studied, 93.6% reported low stress, 3.9% reported moderate stress, and 2.5% reported high stress. Furthermore, similar study was conducted by Putri and Ariana in Indonesian context where the stress level was measured by Student-life Stress Inventory [8]. The stress level was measured by using Student-life Stress Inventory. The Findings show that out of 252 first-year undergraduate students studied, 110 of them (43.6%) reported low stress, 89 respondents reported moderate stress (35.3%), and 53 respondents (21%) reported high stress.

2. Please provide a theoretical foundation of the study.

The literature review was provided on lines 72-124.

Revision:

According to Son, Hegde, Smith, Wang, Sasangohar [9], there were stressors which affected the academic’s life among students during COVID-19 pandemic. These are their own health and the health of loved ones, difficulty in concentration, sleeping habits, social relation/social isolation, academic performance, financial difficulties, depressive thoughts, suicidal thoughts. 

COVID-19 outbreak may enhance fear among the students such as their own health and their loved ones. They were worried about the family and relatives who were more susceptible to the virus such as older adults, those who have existing severe health problems, pregnant women, and women who have just gave birth. Students were also concerned about their family whose jobs increased the risk of exposure to COVID-19 such as social and health care workers. 

There are various changes in terms of educational approaches which potentially lead to difficulty in concentrating on academic work. Son et al. [9] found that most of the students (173, 89%) stated that their home is distractive. For them, home is more appropriate place to than to study (79, 46%). In fact, students were easier to be bothered by their family members at home. Other factors influencing students’ concentration were social media, internet, and video games (19, 11%). Some students (18, 10%) mentioned that online classes were subject to diversion due to social interaction deficiency and continuous attention to computer screen. In addition, monotone life patterns mentioned by some students could negatively affect concentration on academic work (5, 3%).

Students reported that their sleep patterns were disrupted due to COVID-19 pandemic. Further, they mostly stated that they tended to stay up later or wake up later they did before the pandemic. It could be associated with the need to be having online meeting through online platforms which could be carried out anytime and anywhere. Another irritative impact brought by the COVID-19 pandemic was irregular sleep patterns such as inconsistent time to go to bed and to wake up from day to day.

Son et al. [9] stated that most students (167, 86%) answered that the COVID-19 pandemic increased the social isolation. Over half of students (91, 54%) stated that it limited their interactions with friends significantly. Some students (52, 31%) stated that they worried about a lack of in-person interactions. Others (9, 5%) mentioned that disturbance to their outdoor activities have affected their mental health.

Furthermore, Son et al. [9] found that majority of students (159, 82%) concerned on their academic performance impacted by the COVID-19 pandemic. Fear of lower performance and delay in completion of studies are also the reasons to induce stress among students during COVID-19. The biggest challenge was the transition to classes through online platforms (61, 38%). In particular, students concerned on sudden changes in the syllabus, the quality of the classes, technical issues with online applications, and the difficulty of online learning. Some students (36, 23%) worried about progress in research and class projects because of social restrictions to keep social distancing and the lack of physical interactions with other students. Others (23, 14%) stated the uncertainty about their grades on the learning through online platforms to be a major stressor. In addition, others (12, 8%) indicated their lower motivation to learn and tendency to procrastinate.

Majority of students revealed their concerns about the financial situations which were impacted by COVID-19. Some students stated that COVID-19 has affected employment opportunities such as part-time jobs and internships. Others expressed the financial difficulties of their family members, mostly parents, who got laid off and got pay cut during the pandemic.

Regarding the impact of the COVID-19 pandemic, the study by Son et al. [9] found that 44% of students mentioned that they were having some depressive thoughts. Most students attributed the depressive thoughts to factors such as loneliness (33%), insecurity (12%), hopelessness (10%), academic performance concerns (8%), and overthinking (5%).

Out of 195 students, 16 students revealed that the pandemic led to suicidal thoughts with 5% experiencing these thoughts as low level and 3% as moderate level. The reasons were dealing to academic performance, problems with family as they returned home, and fear of insecurity and uncertainty.

3. Please provide a literature review of the study. 

The literature review was provided on lines 125-195.

Revision:

Lazarus and Folkman [10] defined that psychological stress is a relationship between an individual and the environment which is valued by the way people exceed their sources and endanger their well-being. This relationship is going through two significant phases that are cognitive appraisals and coping. The cognitive appraisal is defined as “the process of categorizing an encounter, and its various facets, with respect to its significance for well-being” (Lazarus and Folkman [10], p. 31). Furthermore, cognitive appraisal covers primary appraisal and secondary appraisal. 

Primary appraisal manages to answer “Am I trouble or being benefited, now or in the future, and in what ways?”. When people answer “yes”, it means that the situation can be classified as threat, challenge or loss. Loss suggests harms occurred meanwhile threat and challenge can be meant past experiences. Threat refers to physical and psychological potential while challenge means that someone’s concern on his achievement, reward, and growth; however, this threat and challenge are not correspondent with. Lazarus and Folkman [10] mentioned that threat and challenge appraisals are not two ends of a single continuum. For example, Folkman and Lazarus [10] showed that students waiting for an exam evaluate it as threatening and challenging event. Secondary appraisal refers to an assessment of coping resources. It has an attempt to answer “Can I cope with this situation?”. It reflects someone’s ability to deal with the situation because one has the resources (physical, social, psychological, or material).

Lazarus and Folkman [10] proposed that coping has two main functions. First is to regulate emotions or distresses in the stressful situation (emotion-focused coping). The second one is to manage the problem causing the stress by directly changing the elements of the stressful encounter (problem-focused coping). Even though both are used in most stressful situation, they are nevertheless dependent in terms of the way one appraises the situation and of the antecedents of the model. 

Students who experience stressful conditions will have a negative impact on the performance. For instance, when the students experience prolong stress, the stress may affect their academic achievement and lead into loss of enthusiasm which could be associated with low GPA [11].

Oboth and Odiemo [12] found that that only 35.6% of students experiencing low level of stress while the rest 64.4% were experiencing moderate to high levels of stress. The result showed that stress level and academic performance had a significant relationship. Regression analysis showed that the higher the level of stress, the lower the academic performance among the students. This finding is consistent with other studies which were conducted in Malaysia and Indonesia. In Malaysia, Elias, Ping, Abdullah [13] study which was conducted at Universiti Putra Malaysia, revealed that stress is shown to be significantly correlated to academic performance. On the other hand, in Indonesia, in Tumonggor, Sutanti, Evan, Winata [14] study which was conducted among the students of Krida Wacana Christian University class of 2017 to 2019 during the COVID-19 pandemic, shown that there is a relationship between stress and student’s academic performance.

Furthermore, to define protective factors of stress among the students, some studies were initiated. Majority of researches investigated the relationship between self-efficacy and adversity quotient. Somaratne, Jayawardena, Perera [15] found that level of adversity quotient was significantly related to the level of stress indicating that the higher level of adversity quotient, the lower level of stress. By using Pearson Product-Moment correlation analysis, the results showed that there was a strong negative relationship between perceived stress and all the sub-dimensions of AQ (control, ownership, reach, endurance). By having higher levels of control sub-dimensions, students could be proactively dealing with adverse events and are competent to turn obstacles into opportunity. The negative relationship of ownership sub-dimensions indicates that students with high AQ levels tend to feel liable to improve the obstacles and face them responsibly. Besides that, the negative relationship between reach sub-dimensions and stress indicates that the respondents with high AQ levels do not let obstacles to reach other areas of life that restrict the negative impacts to that particular event. Similarly, as implied by the negative relationship between endurance sub-dimensions and stress, the respondents with high AQ tend to find solutions to overcome the obstacles.

On the other hand, Lee, Kim and Wachholtz [16] found that perceived stress was negatively associated with both self-efficacy among the 279 undergraduate students from three universities in Seoul, and Dae Jeon Korea. In terms of gender distribution, 188 (67.4%) of the participants were female and 91 (32.6%) students were male. It was reported that that self-efficacy and stress among students showed a negative significant relationship indicating also that the higher self-efficacy, the lower stress level of individual. This means if students were confident in their ability to meet the challenges, they would experience positive interpretation of current situation, leading to enhance the chance to avoid having stress.

The rational of conducting current systematic review on the relationship between self-efficacy, adversity quotient, COVID-19-related stress and academic performance is to enhance knowledge on stress among the students during COVID-19 pandemic and to examine the methodological approach in researching student’s stress during the COVID-19 pandemic.

Hence, the present systematic review aims to review research manuscripts during the COVID-19 pandemic which focus on the relationship between self-efficacy, adversity quotient, COVID-19-related stress and academic performance on a range of undergraduate student. Ultimately, based on this systematic review, we will recommend the direction for future research.

Reviewer #1:

1. Your study is quite reasonable, intellectual and informative but in this review article many times every statement or heading start with " This review article", so instead of this statement use suitable word. Over-all this manuscript is accepted.

The similar statements were replaced with suitable words. The changes could be found on lines 37, 39, and 210.

Insertion:

37: The result of the study…

39: Data from published studies will be used…

210: It should cover…

Reviewer #2

1. Include Cochrane library in Information source

We added Cochrane Library source on lines 26, 223, and 314.

Revision: The authors will perform comprehensive searches of published studies in electronic databases such as PMC, PubMed, Scopus, Cochrane Library and Web of Science…

2. Include months with years in Information source

Months and years were added on lines 224 and 232.

Revision: … these are the PubMed Central (PMC), PubMed, Scopus, Cochrane Library and Web of Science databases and published between December 2019 until August 2022.

3. Please provide details list of keywords with Boolean expression.

The keywords for the search were amended on lines 27-29, 225-227 and 230-231.

Revision: The keywords for the search are: ‘self-efficacy’ AND ‘adversity quotient’ AND ‘stress’ AND ‘academic performance’ AND ‘student’ AND ‘COVID-19 pandemic’.

4. Please provide detail and separate heading of data extraction methods.

Data extraction method was added with separated heading. Changes are described on lines 257-263.

Revision: Data extraction

Each accepted manuscript for review will be analyzed through a systematic and careful process. The full text of the articles will be read, exploring their methodology and results. Information on the study’s design, sampling method, sample size, psychological tools used, and operational definitions will be recorded. A risk of bias analysis will be carried out for each individual study. The summary of the PRISMA checklist and a sample for electronic search strategy for the present systematic review will be presented accordingly.

5. Please cite literature properly where necessary such as a systematic narrative synthesis. Cite reference.

The citation was amended on lines 275.

Revision: By using a systematic narrative synthesis [24] …

6. Please provide operational definition of variables.

The operational definition of variables was added on lines 265-274.

Revision: Some key terms are clarified in this study. Firstly, Pajares and Miller [19] assume that self-efficacy refers to student’s beliefs in their ability to master new skills and tasks. Secondly, Adversity Quotient (AQ) is defined as the ability to surmount life’s adversities, and to turn every challenge into opportunities for personal success [20]. Thirdly, stress refers to a person’s physical or emotional response to the demands or pressures of daily life [21]. In academic background, stress pervades the life of students, and tends to affect their mental and physical health, as well as their ability to perform schoolwork effectively [22]. Lastly, academic performance defined by grade point average (GPA). In addition, undergraduate students refer to students at a college or university who has not received a first and especially a bachelor's degree [23].

7. Please explain or define what does undergraduate students means? Detail operational information is mandatory.

The definition of undergraduate students was described on lines 272-274.

Revision: In addition, undergraduate students refer to students at a college or university who has not received a first and especially a bachelor's degree [23].

8. Justify tool using for Bias assessment? Please add Cochrane bias tool or some more tool and compare your findings in actual study.

The tool justification for bias assessment was added on line 285-289. The Cochrane Collaboration’s tool for assessing risk of bias was also added on lines 292-293.

Insertion:

285-289: Systematic reviews aim to collate and synthesise all studies that meet prespecified eligibility criteria using methods that attempt to minimise bias [25]. To get reliable conclusions, review authors must carefully consider that the potential limitations of the included studies (its design, conduct, analysis, and presentation) are appropriate to answer its research question. Many tools for assessing the quality of the study such as the risk of bias assessment tool.

292-293: …and The Cochrane Collaboration’s tool for assessing risk of bias [27].

9. Please discuss the methodology of your proposed study and mention under the heading of discussion?

The Discussion heading was added on lines 308-325.

Revision: Discussion

 The systematic review focuses on the published journal articles discussing the relationship between self-efficacy, adversity quotient, COVID-19-related stress, and academic performance among undergraduate students. It should cover published full-text English articles and studies which examine the undergraduate student population during the COVID-19 pandemic. Five databases will be used, namely the PubMed Central (PMC), PubMed, Scopus, Cochrane Library and Web of Science databases. the articles selected for this study are the articles published between December 2019 until August 2022. 

Reviewers will select articles against the inclusion criteria based on their title, abstract and full-text copies. Each accepted full-text manuscript will be analyzed by using narrative synthesis to explore their methodology such as study design, sampling method, sample size, psychological tools used, and their result on the relationship between self-efficacy, adversity quotient, COVID-19-related stress, and academic performance among the undergraduate students during the COVID-19 pandemic.

To reach the reliable conclusions, the risk of bias of each individual study is determined using two tools. The first is the National Heart, Lung, and Blood Institute Quality Assessment Tool for Observational Cohort and Cross-Sectional Studies and the second is The Cochrane Collaboration’s tool for assessing risk of bias.

10. What will be expected limitation of the study?

The expected limitations were added on lines 326-330.

Revision: Expected limitation

The findings of this review may be restricted by a limitation. The review process inevitably identifies studies that are different in their design, quality of methodology, specific interventions used, also educational background and types of respondents studied. There is potential subjectivity when deciding how similar studies must be.

---

## [Decision Letter · Decision Letter 1]

24 Oct 2022

PONE-D-22-14527R1Relationship between self-efficacy, adversity quotient, covid-19-related stress and academic performance among the undergraduate students: a protocol for a systematic reviewPLOS ONE

Dear Dr. Amit,

Thank you for submitting your manuscript to PLOS ONE. After careful consideration, we feel that it has merit but does not fully meet PLOS ONE’s publication criteria as it currently stands. Therefore, we invite you to submit a revised version of the manuscript that addresses the points raised during the review process.

We look forward to receiving your revised manuscript.

Kind regards,

Muhammad Shahzad Aslam, Ph.D.,M.Phil., Pharm-D

Academic Editor

PLOS ONE

Reviewers' comments:

Reviewer's Responses to Questions

**Comments to the Author**

1. Does the manuscript provide a valid rationale for the proposed study, with clearly identified and justified research questions?

Reviewer #3: Yes

2. Is the protocol technically sound and planned in a manner that will lead to a meaningful outcome and allow testing the stated hypotheses?

Reviewer #3: Yes

3. Is the methodology feasible and described in sufficient detail to allow the work to be replicable?

Reviewer #3: Yes

4. Have the authors described where all data underlying the findings will be made available when the study is complete?

Reviewer #3: No

5. Is the manuscript presented in an intelligible fashion and written in standard English?

Reviewer #3: Yes

6. Review Comments to the Author

You may also provide optional suggestions and comments to authors that they might find helpful in planning their study.

Reviewer #3: Dear Authors,

Thank you for the opportunity to review an interesting article entitled: ‘Relationship between self-efficacy, adversity quotient, covid-19-related stress and academic performance among the undergraduate students: a protocol for a systematic review’. Overall, manuscript is written clearly and sensibly, but the following points should be noted:

[1]. In the abstract at lines 27-29, the search terms used should be in apostrophes, as they are in the text - lines 225-227.

[2]. In lines 67-69 the same information are repeated in two consecutive sentences.

[3]. In lines 84-111, the figures in brackets will be better understood if the authors choose to present the percentages themselves, or replace the comma with a dash - e.g. (173 - 89%).

[4]. In line 314, the sentence should start with a capital letter.

7. PLOS authors have the option to publish the peer review history of their article (what does this mean?). If published, this will include your full peer review and any attached files.

Reviewer #3: **Yes: **Wojciech Marcin Czerski

---

## [Author Response · Author response to Decision Letter 1]

10 Nov 2022

Reviewer #3

1. Have the authors described where all data underlying the findings will be made available when the study is complete? NO.

Response: The description on the availability of data was inserted on lines 319-320.

Revision: The data supporting the findings of this study will be made available within the article and/or its supplementary materials.

2. In the abstract at lines 27-29, the search terms used should be in apostrophes, as they are in the text - lines 225-227.

Response: The apostrophes for search terms were inserted on lines 27-28.

Revision: …‘self-efficacy’ AND ‘adversity quotient’ AND ‘stress’ AND ‘academic performance’ AND ‘student’ AND ‘COVID-19 pandemic’.

3. In lines 67-69 the same information are repeated in two consecutive sentences.

Response: The repeated sentence (i.e., The stress level was measured by using Student-life Stress Inventory) was removed from the paragraph on lines 65-69.

Revision: Furthermore, similar study was conducted by Putri and Ariana in Indonesian context where the stress level was measured by Student-life Stress Inventory [8]. The findings show that out of 252 first-year undergraduate students studied, 110 of them (43.6%) reported low stress, 89 respondents reported moderate stress (35.3%), and 53 respondents (21%) reported high stress.

4. In lines 84-111, the figures in brackets will be better understood if the authors choose to present the percentages themselves, or replace the comma with a dash - e.g. (173 - 89%).

Response: The figures and percentages in brackets were revised as follows.

80-88: There are various changes in terms of educational approaches which potentially lead to difficulty in concentrating on academic work. Son et al. [9] found that most of the students (173 - 89%) stated that their home is distractive. For them, home is more appropriate place to than to study (79 - 46%). In fact, students were easier to be bothered by their family members at home. Other factors influencing students’ concentration were social media, internet, and video games (19 - 11%). Some students (18 - 10%) mentioned that online classes were subject to diversion due to social interaction deficiency and continuous attention to computer screen. In addition, monotone life patterns mentioned by some students could negatively affect concentration on academic work (5 - 3%).

95 – 99: Son et al. [9] stated that most students (167 - 86%) answered that the COVID-19 pandemic increased the social isolation. Over half of students (91 - 54%) stated that it limited their interactions with friends significantly. Some students (52 - 31%) stated that they worried about a lack of in-person interactions. Others (9 - 5%) mentioned that disturbance to their outdoor activities have affected their mental health.

100-110: Furthermore, in 2020, Son et al [9] found that majority of students (159 - 82%) concerned on their academic performance impacted by the COVID-19 pandemic. Fear of lower performance and delay in completion of studies are also the reasons to induce stress among students during COVID-19. The biggest challenge was the transition to classes through online platforms (61 - 38%). In particular, students concerned on sudden changes in the syllabus, the quality of the classes, technical issues with online applications, and the difficulty of online learning. Some students (36 - 23%) worried about progress in research and class projects because of social restrictions to keep social distancing and the lack of physical interactions with other students. Others (23 - 14%) stated the uncertainty about their grades on the learning through online platforms to be a major stressor. In addition, others (12 - 8%) indicated their lower motivation to learn and tendency to procrastinate.

5. In line 314, the sentence should start with a capital letter.

Response: The capital letter was inserted in the revision of this sentence (on lines 312-313).

Revision: The articles which will be selected for this study are the articles published between December 2019 until August 2022.

Additional Review:

1. Please upload a copy of Figure 1 which you refer to in your text on page 11. Or if the figure is no longer to be included as part of the submission please remove all reference to it within the text.

Response: As the guidelines say, which:

• each figure caption should appear directly after the paragraph in which they were first cited, and

• all figures should be uploaded separately as individual files,

Revision: 

• the caption has added on page 11 lines 241 as “Fig 1. PRISMA Flow Chart.”,

• the figure has uploaded and attached as individual files namely “Fig1.tiff”.

---

## [Decision Letter · Decision Letter 2]

21 Nov 2022

Relationship between self-efficacy, adversity quotient, COVID-19-related stress and academic performance among the undergraduate students: a protocol for a systematic review

PONE-D-22-14527R2

Dear,

We’re pleased to inform you that your manuscript has been judged scientifically suitable for publication and will be formally accepted for publication once it meets all outstanding technical requirements.

Kind regards,

Muhammad Shahzad Aslam, Ph.D.,M.Phil., Pharm-D

Academic Editor

PLOS ONE

Additional Editor Comments (optional):

Reviewers' comments:

Reviewer's Responses to Questions

**Comments to the Author**

1. Does the manuscript provide a valid rationale for the proposed study, with clearly identified and justified research questions?

Reviewer #3: Yes

2. Is the protocol technically sound and planned in a manner that will lead to a meaningful outcome and allow testing the stated hypotheses?

Reviewer #3: Yes

3. Is the methodology feasible and described in sufficient detail to allow the work to be replicable?

Reviewer #3: Yes

4. Have the authors described where all data underlying the findings will be made available when the study is complete?

Reviewer #3: Yes

5. Is the manuscript presented in an intelligible fashion and written in standard English?

Reviewer #3: Yes

6. Review Comments to the Author

You may also provide optional suggestions and comments to authors that they might find helpful in planning their study.

Reviewer #3: Dear Authors,

Thank you for the further opportunity to review an interesting article entitled: 'Relationship between self-efficacy, adversity quotient, covid-19-related stress and academic performance among the undergraduate students: a protocol for a systematic review'. All the comments I have indicated have been taken into account.

However, numerical contained in brackets on lines 82-109 will be better readable as separated by semicolons rather than by dashes, as I suggested earlier.

7. PLOS authors have the option to publish the peer review history of their article (what does this mean?). If published, this will include your full peer review and any attached files.

Reviewer #3: **Yes: **Wojciech Marcin Czerski

---

## [Editor Report · Acceptance letter]

23 Nov 2022

PONE-D-22-14527R2 

Relationship between self-efficacy, adversity quotient, COVID-19-related stress and academic performance among the undergraduate students: a protocol for a systematic review 

Dear Dr. Amit:

I'm pleased to inform you that your manuscript has been deemed suitable for publication in PLOS ONE. Congratulations! Your manuscript is now with our production department. 

Kind regards, 

on behalf of

Dr. Muhammad Shahzad Aslam 

Academic Editor

PLOS ONE